# Green Synthesis of Spirooxindoles via Lipase-Catalyzed One-Pot Tandem Reaction in Aqueous Media

Yong Tang [1], Ciduo Wang [1], Hanqing Xie [1], Yuelin Xu [1], Chunyu Wang [2], Chuang Du [3,*], Zhi Wang [1,*] and Lei Wang [1,*]

[1] Key Laboratory of Molecular Enzymology and Engineering of Ministry of Education, School of Life Sciences, Jilin University, Changchun 130023, China
[2] State Key Laboratory of Supramolecular Structure and Materials, Jilin University, Changchun 130023, China
[3] Changchun Institute of Applied Chemistry, Chinese Academy of Sciences, Changchun 130023, China
[*] Correspondence: duch@ciac.ac.cn (C.D.); wangzhi@jlu.edu.cn (Z.W.); w_lei@jlu.edu.cn (L.W.)

**Abstract:** The development of non-natural enzymatic catalysis is important for multicomponent tandem organic transformations. However, the delicate acting environments of biological enzymes still present some challenges in the synthesis of spirooxindole skeleton via enzymatic catalysis. To address these issues, a lipase-catalyzed method was developed for the synthesis of spirooxindole frameworks. Using easily available isatins, cycloketones, and malononitriles as substrates, mild reaction conditions, and a reasonable reaction time, moderate to good yields (67–92%) and excellent functional group tolerance were accomplished via this protocol. The related mechanism explanation is also speculated in this paper.

**Keywords:** spirooxindoles; lipase catalysis; one-pot tandem process; green synthesis

## 1. Introduction

Compared with traditional stepwise processes, one-pot tandem processes represent a greener and more versatile synthesis approach, given their inherent simplicity. The multi-step reaction in one-pot processes can start from relatively simple and easily available raw materials without the separation of intermediates to directly obtain complex molecules, which is evidently economical and environmentally friendly [1]. The one-pot tandem catalytic system combined with chemical catalysis has been widely reported, whereas systems using biocatalysis still have development space and potential [2,3]. As Nicholas Harmer mentioned, sets of enzymes or multiple substrates working together—a "cascade"—to drive desirable reactions in one-pot systems will help deliver the target product on a large scale using fewer resources and generating less waste [4,5]. Large-scale demand for target products, such as fine chemicals, medicine, and food, are present in many fields. The starting reagents used in the cascade process can be derived from low-cost, high-yield isolates, and we can create maximum production value based on the convenience of initial materials.

Spirocytic structures have attracted attention due to their wide range of biological and pharmacological activities [6–8]. Highly functional spirooxindoles have become a research hotspot due to their remarkable and diverse biological activities [9,10]. These oxindole skeleton compounds with various spiral–ring structures have been shown to be used as anti-infective, anti-tumor, and antibacterial materials, and as molecular probes [11–13]. Therefore, spiroxide indole compounds with different structures should be designed and synthesized. At present, various organocatalyzed schemes have been developed to obtain spiroindole structures. In 2010, Perumal's group reported a method for the synthesis of functionalized spirocyclic oxindoles catalyzed by triethylamine [14]. In 2016, Pore's group reported the use of the strong base-DABCO to catalyze the synthesis of functionalized

spirooxindoles [15]. In 2015, Kesavan et al. developed an enantioselective synthesis protocol in toluene, obtaining carbocyclic spirooxindoles with good yield and enantioselectivity using L-proline derived thiourea organocatalyst [16]. In 2021, Parewa et al. manufactured functionalized graphitic carbon nitride ($Sg\text{-}C_3N_4$) and utilized a reusable catalyst for the one-pot production of various spiro-pyrano chromenes and spiro indole-3,10-naphthalene tetracyclic systems in aqueous media [17]. Although these methods can obtain spirooxindoles with good yield in a relatively short time, the reactions need to be catalyzed by weak or strong organic base catalysts, and some catalysts require complex synthesis procedures. For example, thiourea catalysts reported in Kesavan's work require chiral prolines for derivational synthesis [16], and the catalyst reported by Parewa requires high temperature, a strong acid solution and tedious characterization [17]. In addition, most of the methods reported need to be carried out in organic solvents, such as toluene or ethylene glycol; these organic solvents do not conform to the concept of green chemistry to some extent (Figure 1).

**Previous works** (Reprinted/adapted with permission from Ref. [14-17]) :

Perumal's work [14] [Paramasivan T. Perumal, 2010] :

Pore's work [15] [Dattaprasad M. Pore, 2016] :

Kesavan's work [16] [Venkitasamy Kesavan, 2015] :

Parewa's work [17] [Vijay Parewa, 2021] :

**This work:**

**Figure 1.** Overview of methods for synthesizing spiroxide indole compounds.

With the introduction of the concept of green chemistry and the emergence of new technologies, biocatalysts are gradually becoming an important part of the field of chemical

synthesis. In many cases, biocatalysts have replaced traditional chemical catalysts and have significant application prospects in the future [18–23]. Through enzymatic reactions, we can construct drug modules and biologic therapeutics effectively, and reduce the use of polluting chemicals and solvents [24–27]. The synthesis of spirooxindole compounds catalyzed by enzymes has been partially studied. In 2011, Zhang reported the synthesis of spirooxindole catalyzed via lipase (porcine pancreatic lipase), but the process was carried out in a mixed solvent of ethanol/water, and the ratio of water significantly affected the yield [28]. In 2014, Lin's group used isatin derivatives and 1,3-dicarbonyl as starting materials; spirooxindole derivatives were obtained via a two-component enzymatic (Acylase Amano from Aspergillus oryzae) catalytic process in ethylene glycol [29]. However, the enzymatic synthesis of spirooxindole is still a hot spot in enzyme catalysis. Lipase, as an efficient and green biocatalyst, has been widely studied by researchers due to its good applicability of reaction types, stable temperature, solvent compatibility and commercial simplicity [30–32].

In this study, we investigated the synthesis of spirooxindoles from isatin, malononitrile and cycloketone substrates with lipase from porcine pancreatic lipase (PPL) in the presence of water to expand the study of non-natural enzymatic reactions and explore favorable biocatalytic synthesis. To the best of our knowledge, the synthesis of spirooxindole using a one-pot tandem process catalyzed via lipase in aqueous media has not been reported before (Figure 1).

## 2. Results and Discussion

### 2.1. Optimization of Reaction Conditions for the One-Pot Tandem Process

**1a**, **2a,** and **3** were selected as model substrates to optimize a series of reaction conditions. We discovered the catalytic effects of different lipases at a temperature of 40 °C (Table 1), and selected several lipases that showed different catalytic activities. Among these lipases, porcine pancreatic lipase (PPL) showed the highest reactivity (Table 1, entry 1 versus entries 2–5). Although PSL, BSA, CALB, and Novozym 435 had certain catalytic effects, the target product was obtained with low yield (Table 1, entries 2–5), whereas **4a** was not obtained when denatured PPL was replaced with PPL (Table 1, entry 6); similar results occurred in the controlled experiment (Table 1, entry 7). The above results indicated that the difference in the lipase active site was crucial for the catalytic synthesis of spirooxindoles. Based on the highest lipase activity temperature range reported in the relevant literature [33], analyses were conducted regarding experimental temperature (see Supplementary Materials, Table S1).

Solvents could significantly affect the activity of enzymes in the catalytic enzymatic reaction. Therefore, we proceeded to examine the effects of solvents. The model reaction showed excellent reactivity in the presence of polar solvents, such as EtOH, DMF, DMSO, and water (Table 2, entries 1–4). However, when THF, toluene, EA, and DCM were used as reaction solvents separately, the catalytic effect of PPL was extremely poor or even non-reactive (Table 2, entries 5–8). Although PPL in DMF or DMSO could make the model reaction achieve the highest yield in a relatively short time (Table 2, entries 2–3), water, as the best medium in enzymatic chemical reactions, has a series of advantages, including its safety and non-toxicity, and that it is a green solvent [34]. In addition, in subsequent separation and purification processes, due to the low solubility of the target product **4a** in the water phase, we could easily obtain the target product via simple filtration, washing, and drying, rather than using a complicated column chromatography separation. Even though the reaction takes longer in the aqueous phase, we still chose water as the best solvent for an almost identical yield.

**Table 1.** Optimization of lipase types for the one-pot tandem synthesis of **4a**.

| Entry | Lipase [1] | Yield (%) [2] |
|-------|-----------|---------------|
| 1 | PPL | 88 |
| 2 | PSL | 40 |
| 3 | BSA | 45 |
| 4 | CALB | 35 |
| 5 | Novozym 435 | 38 |
| 6 | PPL [3] | N.D [4] |
| 7 | Control | N.D [4] |

Reaction conditions: **1a** (0.2 mmol), **2a** (0.2 mmol), **3** (0.6 mmol), lipase (15 mg), water (3 mL), 40 °C, 72 h. [1] PPL (porcine pancreatic lipase); PSL (lipase from *Pseudomonas* sp.); BSA (albumin from *bovine serum*); CALB (*C. antarctica* lipase B); Novozym 435 (a commercial immobilized lipase B from *C. antarctica*). [2] Isolated yield. [3] The denatured PPL was obtained by heating PPL to 100 °C for 12 h in water. [4] Not detected.

**Table 2.** Optimization of solvent for the one-pot tandem synthesis of **4a**.

| Entry | Solvent [1] | Yield (%) [2] |
|-------|-------------|---------------|
| 1 | EtOH | 86 |
| 2 | DMF [3] | 90 |
| 3 | DMSO [3] | 90 |
| 4 | $H_2O$ | 88 |
| 5 | THF | 35 |
| 6 | Toluene | 23 |
| 7 | EA | N.D [4] |
| 8 | DCM | N.D [4] |

Reaction conditions: **1a** (0.2 mmol), **2a** (0.2 mmol), **3** (0.6 mmol), solvent (3 mL), PPL (15 mg), 40 °C, 72 h. [1] EtOH (ethanol), DMF (*N,N*–dimethylformamide), DMSO (dimethyl sulfoxide), $H_2O$ (water), THF (tetrahydrofuran), EA (ethyl acetate), DCM (dichloromethane). [2] Isolated yield. [3] Reaction for 3 h. [4] Not detected.

The enzyme dosage also has an evident effect on the reaction. Insufficient enzyme reduces catalytic efficiency, whereas excessive enzyme increases production cost, and even leads to negative effects to some extent. Therefore, we explored the influence of enzyme dosage. Figure 2 shows that the yield of spirooxindole **4a** increased as the lipase dosage increased from 5 mg to 15 mg, but the yield of **4a** decreased as the lipase dosage increased from 20 mg to 25 mg. This phenomenon might have been caused by excessive enzyme aggregation in a quantitative solvent that was not conducive to the contact between substrates and enzyme active centers. Thus, 15 mg PPL was the optimal reaction dosage.

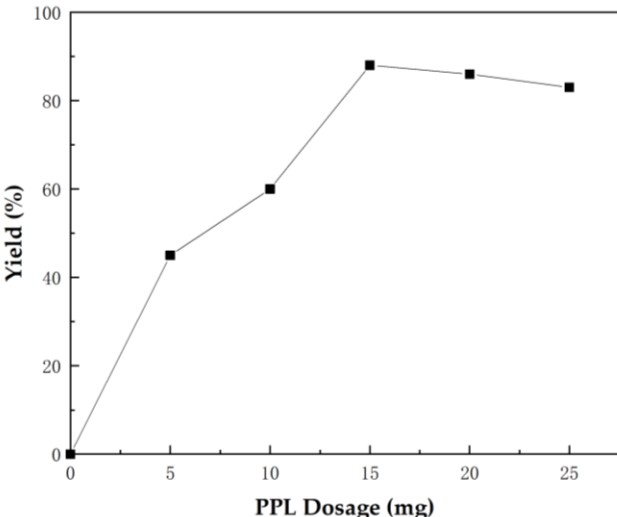

**Figure 2.** Optimization of PPL dosage for the one-pot tandem synthesis of **4a**. Reaction conditions: **1a** (0.2 mmol), **2a** (0.2 mmol), **3** (0.6 mmol), $H_2O$ (3 mL), 40 °C for 72 h.

### 2.2. Substrate Scopes for the One-Pot Tandem Process

With optimized conditions in hand, respective series of isatins **1** and cyclic ketones **2** were examined using this one-pot tandem process; results are summarized in Figure 3. Cyclohexanone **2a** easily reacted with isatins at different substitution sites and afforded the final products **4a–4m** with 67–90% isolated yields. It was notable that their electronic properties did not significantly affect their yields. One-pot tandem reactions with substrates (**2a–2m**) had electron-withdrawing and electron-donating substituents at 2- to 7-positions. To be clear, all products except **4c** (67% isolated yield) achieved excellent yield, which might have been caused by the high steric hindrance of the 4-substituted site. In addition, 6-membered cycloketone substrates were tested for this transformation. Similarly, **1a** smoothly reacted with **2n–2r**, producing desired products **4n–4r** in high isolated yields of 84–92%.

### 2.3. Mechanism of the One-Pot Tandem Process

To understand the mechanism better, some control experiments were performed. We found that both isatin **1** and cyclohexanone **2** could form corresponding intermediates **Ia** and **IIa** with malononitrile **3** in either a single system or a mixed system, indicating that the formation of intermediates **Ia** and **IIa** in a one-pot tandem system was a spontaneous Knoevenagel condensation process (Figure 4A,B versus Figure 4C). Subsequently, it was found that the reaction between intermediate **Ia** and **IIa** could not form the target product **4a** without the action of PPL. On the contrary, **4a** could be effectively generated under the catalysis of PPL (Figure 4D versus Figure 4E).

On the basis of our experimental results, we speculated regarding the plausible mechanism of this PPL-catalyzed one-pot tandem reaction (Figure 5). Given the strong electron-withdrawing ability of –CN, first, using this one-pot tandem process, adducts alkenyl dinitrile **I** and **II** were obtained via Knoevenagel condensation of isatin **1** and cycloketone **2**, respectively, using malononitrile **3**. Next, the adduct **II** was deprotonated using Asp-His dyad to produce intermediate **III**. Immediately after that, the carboanion of the intermediate **III** nucleophile attacked the C=C of the isatilidenemalononitrile **I** and formed **IV**, which further attacked the cyanogroup and cyclized to obtain the anionic intermediate **V**. Finally, **V** was protonated/deprotonated to provide target product **4;** the regeneration of PPL maintained the catalytic cycle.

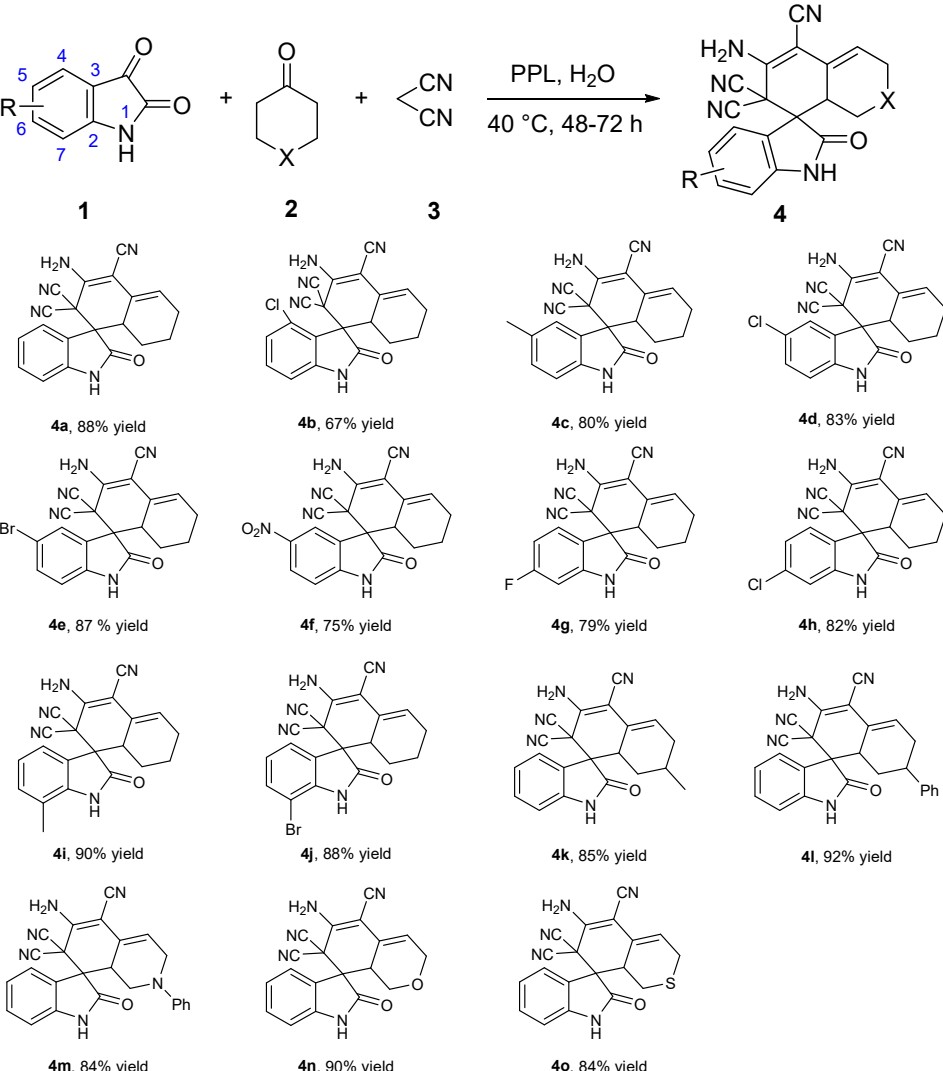

**Figure 3.** Substrate scope for the one-pot tandem synthesis of **4**. Reaction conditions: **1** (0.2 mmol), **2** (0.2 mmol), **3** (0.6 mmol), PPL(15 mg), $H_2O$ (3 mL), 40 °C; **4a–4l** for 72 h, and **4m–4o** for 48 h. Yields are the isolated yield.

**Spontaneous Knoevenagel condensation**

**Figure 4.** *Cont.*

**The reaction process catalyzed by PPL**

**Figure 4.** Control experiments. (**A**) and (**B**): spontaneous Knoevenagle condensation in single system; (**C**): spontaneous Knoevenagle condensation in mixed system; (**D**) and (**E**): Investigation on the catalytic effect of PPL. [1]H NMR of intermediates **Ia** (400 MHz, DMSO-*d*6) δ 11.23 (s, 1H), 7.90 (d, *J* = 7.8 Hz, 1H), 7.59 (td, *J* = 7.8, 1.2 Hz, 1H), 7.16 (td, *J* = 7.7, 1.0 Hz, 1H), 6.96 (d, *J* = 7.9 Hz, 1H). [1]H NMR of intermediates **IIa** (400 MHz, DMSO-*d*6) δ 2.09–2.21 (m, 4H), 1.66–1.49 (m, 6H).

**Figure 5.** Proposed reaction mechanism for the synthesis of spirooxindoles. I and II: alkenyl dinitrile adducts; III: vinyl dinitrile carboanion; IV: nucleophilic addition intermediate; V: cyclization intermediate of carbanion.

## 3. Materials and Methods

### 3.1. General Information

PPL (porcine pancreas lipase), PSL (*Pseudomonas* sp. lipase), and CalB (lipase B from *Candida antarctica*) were purchased from Shanghai Yuan Ye Biological Technology Company (Shanghai, China), and Novozym 435 was purchased from Sigma–Aldrich China Co. (Beijing, China). All other chemical reagents were purchased from commercial suppliers (Bide Pharmatech (Shanghai, China), Aladdin (Shanghai, China), and Energy Chemical (Beijing, China)). All commercially available reagents and solvents were used without further purification. Proton nuclear magnetic resonance ($^1$H NMR) spectra were recorded using a 400 MHz spectrometer in DMSO. Chemical shifts of protons were reported in parts per million downfield from tetramethyl silane (TMS) and referenced to residual protium in the NMR solvent (DMSO = δ 2.50 ppm). NMR data are presented as follows: chemical shift (δ ppm), multiplicity (s = singlet, d = doublet, t = triplet, q = quartet, m = multiplet, and br = broad), coupling constant in Hertz (Hz), and integration. A 100 MHz spectrometer in DMSO (δ 39.52 ppm) was used to report $^{13}$C NMR spectra. Experiments were performed in triplicate, and all data were obtained based on the average values.

### 3.2. General Procedure for Lipase-Catalyzed Synthesis of **4**

PPL (15 mg) was added into a mixture of isatins (**1**, 0.2 mmol), cycloketones (**2**, 0.2 mmol), and malononitrile (**3**, 0.6 mmol) in water (3 mL) at 40 °C for 48–72 h. After the reaction was complete, as indicated by TLC, the precipitate was filtered and washed using 20% ethanol/petroleum ether, then dried in a vacuum to afford pure and solid product (**4**). All the isolated products were well characterized by their NMR.

## 4. Conclusions

In conclusion, we have successfully established a novel lipase catalyzed one-pot tandem reaction to synthesize a series of spirooxindoles. This method used lipase to catalyze the reaction of three components in aqueous solution. The developed three-component enzymatic process requires only common equipment and obtainable raw materials. Simple filtration and washing operations result in product with an excellent yield, without any further purification steps. Combined with the green chemistry concept that is now being advocated, this non-natural enzymatic method for synthesizing spirooxindole structures is expected to be widely used in the future. Furthermore, immobilization is an efficient tool for improving enzyme features in the biotechnology toolbox. Through enzyme immobilization, many enzyme limitations can be addressed; for example, enzyme stability can be improved; enzyme selectivity, specificity, and activity can be altered and inhibitions reduced; and the operation window and resistance to chemicals can be enlarged, and even be coupled to enzyme purification [35–39]. Therefore, to improve the feasibility and efficiency of this synthetic method, related PPL immobilization methods are being studied and will be reported in due course.

**Supplementary Materials:** The following supporting information can be downloaded at https://www.mdpi.com/article/10.3390/catal13010143/s1, Figure S1: Data of Products; Figure S2: Spectra of Products; Table S1: Optimization of temperature for the one-pot tandem synthesis of **4a**.

**Author Contributions:** Investigation, methodology, visualization, writing—original draft, and formal analysis, Y.T.; methodology, C.W. (Ciduo Wang); visualization, H.X.; formal analysis, Y.T., C.W. (Chunyu Wang); H.X., Y.X. and C.D.; supervision, conceptualization, funding acquisition, and writing review and editing, Z.W. and L.W. All authors have read and agreed to the published version of the manuscript.

**Funding:** This research was funded by the Science and Technology Development Program of Jilin Province (No. 20200301029RQ and No. 20200201396JC) and the Key Research Development Program of Jilin Province (No. 20200402067NC).

**Data Availability Statement:** Data presented in this study are available in the Supplementary Materials.

**Acknowledgments:** We gratefully acknowledge the Science and Technology Development Program of Jilin Province (No. 20200301029RQ and No. 20200201396JC) and the Key Research Development Program of Jilin Province (No. 20200402067NC).

**Conflicts of Interest:** The authors declare no conflict of interest.

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
