# Peer review of "Green Synthesis of Spirooxindoles via Lipase-Catalyzed One-Pot Tandem Reaction in Aqueous Media"

_catalysts, doi:10.3390/catal13010143_

Round 1
Reviewer 1 Report
Manuscript by Wang et al. reports the synthesis of spirooxindoles via enzymatic catalysis under aqueous medium. The method is greener compared to the previous reports. The present protocol is synthetically attractive. The substrate scope of isatin is studied well, however substrate scope is not investigated with active methylene compounds other than malononitrile. In supporting information most of proton spectra are good quality but why they did not provide 13C spectra and HRMS? I would recommend this work for publication only after major revision.
1. The authors are encouraged to probe whether active methylene compounds other than malononitrile (ethylcyanoacetate, ethylnitroacetate, benzoylacetonitrile) could be used in the reactions.
2. To support the mechanism of reaction, they should provide some control experiments in which they would detect any of intermediates of Figure 3 by using GCMS or LCMS or by NMR.
3. Author should provide 13C spectra and HRMS of all compounds.
Author Response
Response to Reviewer 1 Comments
Manuscript by Wang et al. reports the synthesis of spirooxindoles via enzymatic catalysis under aqueous medium. The method is greener compared to the previous reports. The present protocol is synthetically attractive. The substrate scope of isatin is studied well, however substrate scope is not investigated with active methylene compounds other than malononitrile. In supporting information most of proton spectra are good quality but why they did not provide 13C spectra and HRMS? I would recommend this work for publication only after major revision.
Point 1: The authors are encouraged to probe whether active methylene compounds other than malononitrile (ethylcyanoacetate, ethylnitroacetate, benzoylacetonitrile) could be used in the reactions.
Response 1: Thanks for the helpful suggestion. At present, our university has been closed in advance due to the impact of the COVID-19, and experiments of active methylene compound you mentioned can not be conducted. We will report them in the future and we sincerely appreciate your understanding and help!
Point 2: To support the mechanism of reaction, they should provide some control experiments in which they would detect any of intermediates of Figure 3 by using GCMS or LCMS or by NMR.
Response 2: Thanks for the helpful suggestion. We have added detailed data and descriptions in the revised MS.
Point 3: Author should provide 13C spectra and HRMS of all compounds.
Response 3: Thanks for the helpful suggestion. We have added detailed data in the revised SI.
Reviewer 2 Report
This manuscript Catalyst-2122088 is a fine addition to the literature in the development of novel biocatalytic methods for the production of key drugs in mild conditions. The authors provide a new synthetic pathway with moderate to high yields of spirooxindoles, key intermediates to novel cancer drugs, from available and relatively simple chemicals using enzymes as catalysts. The paper is short, to the point, really well written...it is a pleasure to read it.
There are some aspects that could be improved:
1) Intro: please, define the needs and the availability at large scale in the introduction: what is the need for the targeted product? what is the production of the starting reagents?
2) Intro: please give some examples of complex catalysts and dangerous organic solvents used in other synthetic approaches.
3) Intro: Can the authors comment on the results of the following authors? Their targeted compounds and the procedures seem relatively similar to those proposed in this work, clearly defining differences with the synthetic route developed by the authors.
Chai, S. J., Lai, Y. F., Xu, J. C., Zheng, H., Zhu, Q., & Zhang, P. F. (2011). One‐pot synthesis of spirooxindole derivatives catalyzed by lipase in the presence of water. Advanced Synthesis & Catalysis, 353(2‐3), 371-375.
Liang, Y. R., Chen, X. Y., Wu, Q., & Lin, X. F. (2015). Diastereoselective synthesis of spirooxindole derivatives via biocatalytic domino reaction. Tetrahedron, 71(4), 616-621.
4) Results and discussion, end of page 4: The authors can also comment on the advantages of water for the purification of the targeted products.
5) Results and discussion, page 5, just before figure 2: Have the authors considered the possibility of enzyme immobilization to recover it and reduce purification costs and complexity? They can extend slightly the discussion in this regard.
6) Please, provide Table 3.
7) Results and discussion, page 6: about the plausible mechanism, is there any evidence observed by the authors that supports the mechanism speculation?
For minor aspects, please see the attached pdf.

Author Response
Response to Reviewer 2 Comments
This manuscript Catalyst-2122088 is a fine addition to the literature in the development of novel biocatalytic methods for the production of key drugs in mild conditions. The authors provide a new synthetic pathway with moderate to high yields of spirooxindoles, key intermediates to novel cancer drugs, from available and relatively simple chemicals using enzymes as catalysts. The paper is short, to the point, really well written...it is a pleasure to read it.
There are some aspects that could be improved:
Point 1: Intro: please, define the needs and the availability at large scale in the introduction: what is the need for the targeted product? what is the production of the starting reagents?
Response 1: Thanks for the helpful suggestion. We added detailed discriptions in the revised MS.
Point 2: Intro: please give some examples of complex catalysts and dangerous organic solvents used in other synthetic approaches.
Response 2: Thanks for the helpful suggestion. We added detailed discriptions in the revised MS.
Point 3: Intro: Can the authors comment on the results of the following authors? Their targeted compounds and the procedures seem relatively similar to those proposed in this work, clearly defining differences with the synthetic route developed by the authors.
Chai, S. J., Lai, Y. F., Xu, J. C., Zheng, H., Zhu, Q., & Zhang, P. F. (2011). One‐pot synthesis of spirooxindole derivatives catalyzed by lipase in the presence of water. Advanced Synthesis & Catalysis, 353(2‐3), 371-375.
Liang, Y. R., Chen, X. Y., Wu, Q., & Lin, X. F. (2015). Diastereoselective synthesis of spirooxindole derivatives via biocatalytic domino reaction. Tetrahedron, 71(4), 616-621.
Response 3: Thanks for the helpful suggestion. In Zhang’s work (2011), the synthesis of spiroxindole was carried out in a mixed solvent of ethanol/water, and the ratio of water significantly affected the yield. In Lin’s work (2015), they used isatin derivatives and 1,3-dicarbonyl as starting materials, spiroxindole derivatives were obtained by a two-component enzymatic (Acylase Amano from Aspergillus oryzae) catalytic process in ethylene glycol. In addition, target products need to be separated by column chromatography. In our report, we only use water as a solvent, a series of spiroxindole compounds were obtained by lipase (PPL) catalyzed one-pot tandem reaction of three components, which accords with the concept of green chemistry. We have added detailed descriptions in the revised MS.
Round 2
Reviewer 1 Report
Manuscript has been sufficiently improved for publication. I recommend the manuscript for publication in Catalysts.